# $X$-Shot : A Single System to Handle Frequent, Few-shot and Zero-shot Labels in Classification

## Abstract

In recent years, few-shot and zero-shot learning, which focus on labels with limited annotated instances, have garnered significant attention. Traditional approaches often treat freq-shot (labels with numerous instances), few-shot, and zero-shot learning as distinct challenges, optimizing systems for just one of these scenarios. Yet, in real-world settings, label occurrences vary greatly. Some labels might appear thousands of times, while others might only appear sporadically or not at all. Ideally, a system should accommodate any label, regardless of its training frequency. Notably, while few-shot systems often falter on zero-shot tasks, zero-shot systems don't leverage available annotations when certain downstream labels possess them. For practical deployment, it's crucial that a system can adapt to any label occurrence. We introduce a novel classification challenge: $X$-Shot , reflecting a real-world context where freq-shot, few-shot, and zero-shot labels emerge without predefined limits. Here, $X$ can span from 0 to $+\infty$. The crux of $X$-Shot centers on open-domain generalization and devising a system versatile enough to manage various label scenarios. Our solution leverages Instruction Learning, bolstered by data autonomously generated by pre-trained Language Models (PLMs). Our unified system, $X$-Shot, surpasses preceding state-of-the-art techniques on three benchmark datasets across diverse domains in both single-label and multi-label classifications. This is the first work addressing $X$-Shot learning, where $X$ remains variable.[1]

## 1 Introduction

Over recent years, few-shot and zero-shot learning techniques have seen significant advancements, aiming to address the challenge of training models with scant or even no annotated instances for specific labels (Bragg et al., 2021; Xia et al., 2020). Historically, the fields of frequent-shot, few-shot, and zero-shot learning have been approached as distinct paradigms, with systems optimized uniquely for each setting. Yet, in real-world scenarios, label frequencies can exhibit broad variation, with certain labels occurring prolifically, and others being scarce or completely absent. Given this variability, it becomes imperative to craft learning systems adept at managing labels across the full frequency spectrum. Regrettably, current few-shot systems often fall short when confronted with zero-shot challenges(Zhang et al., 2022; Cui et al., 2022; Zhao et al., 2021). In contrast, zero-shot systems, while adept in their domain, typically overlook the potential benefits of available annotations(Zhang et al., 2019; Obamuyide & Vlachos, 2018; Yin et al., 2019). Thus, mastering the ability to handle all conceivable label occurrences is paramount for systems aiming for practical deployment.

In this paper, we introduce an innovative and inherently more challenging task, termed $X$-Shot . This task mirrors real-world environments where label frequencies span a continuum, seamlessly incorporating frequent-shot, few-shot, and zero-shot instances, all without a priori constraints. In this paradigm, the variable $X$ is unbounded, ranging freely within the interval $[0, +\infty)$. At the heart of $X$-Shot lies the objective of attaining open-domain generalization and architecting a system resilient across a plethora of label scenarios.

---

[1]Data & code will be released upon acceptance.

Tackling $X$-Shot spawns two core technical conundrums: ($\mathcal{Q}_1$) Amidst the paucity of annotations characteristic of few-shot and zero-shot contexts, how might one identify apt sources of indirect supervision (Yin et al., 2023) to navigate the $X$-Shot setting? ($\mathcal{Q}_2$) Traditional multi-class classifiers grapple with the heterogeneity of label sizes across tasks, often mandating distinct classification heads tailored to these variations. Here, the challenge is formulating a cohesive system capable of effectively managing labels of diverse sizes.

To address $\mathcal{Q}_1$, we tap into the availability of indirect supervision from instruction tuning datasets, such as Super-Naturalinstructions (Wang et al., 2022). These datasets primarily contain various NLP tasks enriched with textual instructions. Our method involves pretraining our model on these datasets, aiming for robust generalization to the unseen $X$-Shot task when supplemented with pertinent instructions. For ($\mathcal{Q}_2$), we advocate a triplet-oriented binary classifier. This classifier functions by accepting a triplet of (`instruction`, `input`, `label`), anticipating a binary response (Yes" or No") that confirms the suitability of the `label` for the specified `input` under the given `instruction`. Such a triplet-oriented classifier acts as a cohesive architecture, adept at managing text classification tasks with labels of varied dimensions. By amalgamating solutions for both $\mathcal{Q}_1$ and $\mathcal{Q}_2$, we forge a holistic framework, $X$-Shot. This framework capitalizes on indirect supervision sourced from a diverse set of tasks, incorporating instructions as guidance, and thus presents a unified architecture proficient in handling text classification challenges with both open-shot and open-size labels.

No existing datasets explicitly cater to this challenge. To evaluate our system, we turn to three representative classification tasks: relation classification, ultra-fine entity typing, and situation detection. We reconfigure their associated datasets: *FewRel* (Han et al., 2018), *UFET* (Choi et al., 2018), and *Situation* (Yin et al., 2019) to simultaneously encapsulate frequent-shot, few-shot, and zero-shot instances. Sourced from diverse domains (Wikipedia, crowdsourcing, and more), and featuring vast label counts (ranging from 12 to the thousands), these datasets pose a formidable challenge to contemporary text classification systems. Moreover, both *UFET* and *Situation* function as multi-label classification datasets. The *Situation* dataset uniquely integrates an " None" label, further amplifying the realistic nature of the task. Empirical results reveal our system's resilience across datasets and instruction templates, consistently outclassing leading methods, including GPT, in frequent-shot, few-shot, and zero-shot contexts.

Our contributions can be summarized as follows: (i) We introduce $X$-Shot, a hitherto underexplored, open-domain open-shot text classification problem that mirrors real-world complexities. (ii) We innovate a unique problem setting that reframes any text classification challenge into a binary classification task, adaptable to any number of labels and occurrences. (iii) Our $X$-Shot, harnessing the potential of Instruction Tuning datasets, excels past existing approaches, demonstrating versatility across various domains, label magnitudes, and classification paradigms.

## 2 RELATED WORK

**Few-shot Learning.** Few-shot learning refers to machine learning methods that can perform tasks with only a few labeled training examples. This technique has gained traction in NLP for two reasons: (i) labeled data can be expensive to obtain and (ii) extensive training or fine-tuning, particularly with large models, can be both costly and unstable. Ideally, a model would generalize from a handful of examples, capturing the core knowledge. The main challenge lies in effectively using limited labeled samples for broad generalizations. Initially, the approach to few-shot learning was metric-based, focusing on a shared feature space and distance metrics for label predictions (Vinyals et al., 2016; Snell et al., 2017; Sung et al., 2018). Recently, Large Language Models (LLMs) have been recognized as efficient few-shot learners. Fine-tuning these pre-trained LLMs with minimal samples often produces notable results (Brown et al., 2020). Additionally, due to the success of prompting in GPT models, prompt-tuning has been applied to tackle classification problems under few-shot settings (Zhang et al., 2022; Cui et al., 2022; Zhao et al., 2021). However, these methods don't typically manage zero-shot scenarios where certain labels are without annotated data.

**Zero-shot Learning.** Building on the concept of few-shot learning, we transition to the even more challenging zero-shot learning where no labeled examples are available. Early techniques in this domain employed metrics to align texts and labels in shared spaces. More recent works adopted

word embeddings from pre-trained language models to represent the meaning of the text or the label. The latest work enhanced the embedding representations by integrating class descriptions, class hierarchy, and the word-to-label paths found within ConceptNet(Zhang et al., 2019). Today's LLMs are so adept that they can tackle NLP tasks without any labeled instances, either by reformatting the classification tasks or through in-context learning as seen with the GPT models (Brown et al., 2020; Wei et al., 2022). Similarly, an alternative approach is to calibrate and score outputs from LLM models for the label assignment(Zhao et al., 2021; Holtzman et al., 2021; Min et al., 2022). The most recent trend in zero-shot text classification is to draw on the power of indirect supervision from other well-annotated NLP tasks, like text entailment (Obamuyide & Vlachos, 2018; Yin et al., 2019). Still, these methods don't fully utilize annotations when they exist for labels.

**Indirect Supervision** There's a burgeoning interest in indirect supervision. Here, easily available signals from relevant tasks are used to aid in learning the target task, especially when task-specific supervision is in short supply. The technique of using entailment for indirect supervision in zero-shot classification was pioneered by (Yin et al., 2019) and has since been adapted for a variety of NLP tasks, including few-shot intent identification (Zhang et al., 2020), event argument extraction (Sainz et al., 2022), and relation extraction (Xia et al., 2021). Beyond entailment, knowledge from areas like question answering (Yin et al., 2021) and summarization (Lu et al., 2022) has been incorporated. Recent studies have demonstrated that modern language models, after fine-tuning on a plethora of instruction-based tasks, can generalize to multiple unseen tasks (Wang et al., 2022; Mishra et al., 2022; Ye et al., 2021). Our work is inspired by the observed efficacy of NLP models when given task instructions and their ability to generalize knowledge across tasks.

## 3 PROBLEM STATEMENT

$X$-Shot has the following requirements:

• **Input** $t$: Versatile text in form, length, and domain.

• **Label space** $L$: $L$ contains arbitrary size of labels: $\{\cdots, l_i, \cdots\}$ and an optional *None* label (i.e., all labels in $L$ are incorrect for the input). Within $L$, some labels are zero-shot, some are few-shot, and some are frequent.

Then, the task of $X$-Shot is to figure out a subset of $L_s \in L$ that are correct for the input $t$, where $|L_s|$ can be zero (i.e., "*None*"), 1 or >1.

**Research questions of $X$-Shot:** i) Given that the above formulation encompasses various text classification problems, how can we move away from constructing individual models for each problem, and instead develop a singular classifier adept at handling diverse classification challenges? ii) Beyond frequently-encountered labels, low-shot labels necessitate additional supervision for effective reasoning. Where can we source this supervision? In the following section, we delve deeper into our approach concerning the universal system and the process of seeking supervision.

## 4 METHODOLOGY

This section outlines our approach to the $X$-Shot problem. We first explain our process of transforming all classification problems into a unified binary classification framework. Next, we discuss the type of supervision we gather to address this problem with limited annotations.

### 4.1 SYSTEM ARCHITECTURE FOR $X$-SHOT

We've devised a broad architecture that seamlessly transitions most classification challenges into a unified, instruction-driven binary classification task. As depicted in Figure 1, for any text classification task with its set of inputs and labels, we model it as (instruction, input, label) triplet. The task then becomes determining if the label is appropriate ("Yes") or not ("No") for the input given an instruction. This new framework is referred to as $X$-Shot.

$X$-Shot can capably manage both multi-class and multi-label classification challenges. Instead of converting labels into numerical IDs as traditional supervised classifiers do, we retain the actual

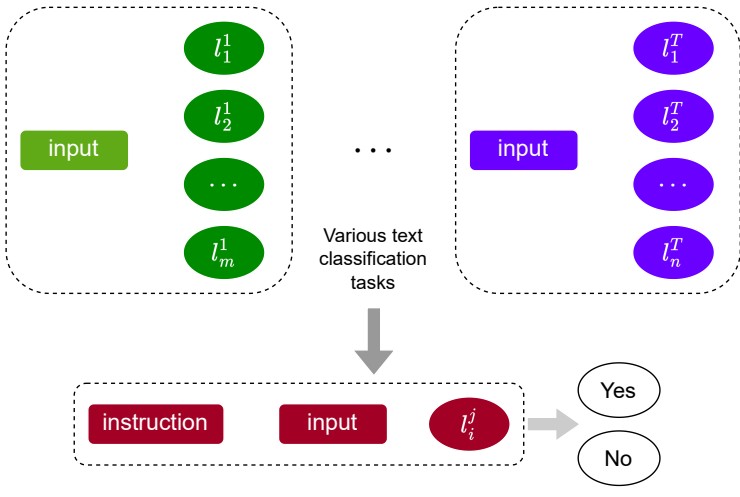

Figure 1: Our $X$-Shot unifies various text classification tasks as an instruction tuning problem.

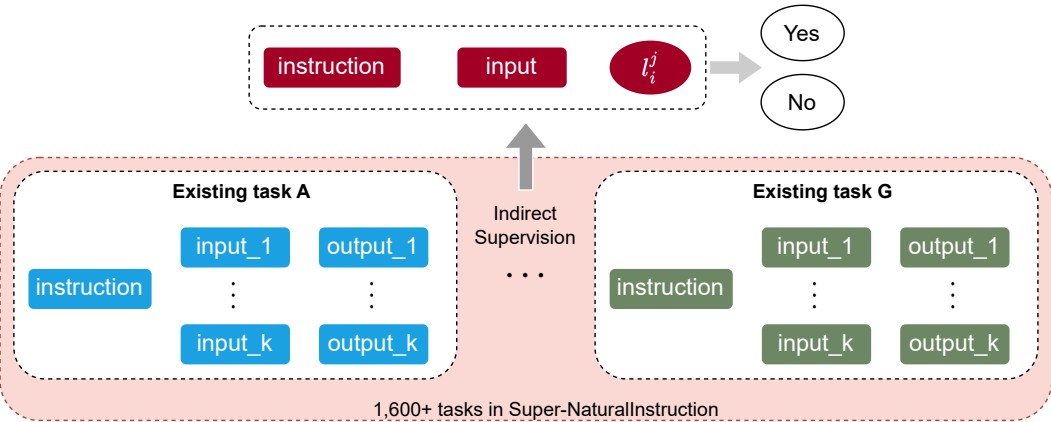

Figure 2: Indirect supervision for $X$-Shot.

label names. Optionally, we can also employ sophisticated verbalizers (Schick & Schütze, 2021) to enhance the expression of the label. This ensures a more intuitive understanding of the relationship between inputs and labels, all within the context of task instructions.

$X$-Shot paves the way to tackle a variety of low-shot text classification tasks using an instruction-guided approach. Two primary challenges arise: i) Ensuring the model comprehends the instructions, and ii) Guiding the model to identify seldom seen or entirely new labels. We'll delve deeper into our supervision-seeking approaches to address these challenges in the following subsection

### 4.2 SUPERVISION ACQUISITION FOR LOW-SHOT LABELS

In this section, we will introduce how we conduct and combine *Indirect Supervision* and *GPT supervision* to solve $X$-Shot .

**Indirect Supervision.**    Previous best-performing systems for low-shot text classification have primarily relied on indirect supervision *from a single source task*. Examples of these source tasks include natural language inference (Yin et al., 2019) and summarization (Lu et al., 2022). This approach presents three main drawbacks: i) the usable supervision from the single source task is finite, and there's often a domain mismatch between the source task and the target classification tasks; ii) typically, instances of the target problems need to be reformatted to align with the specific source tasks to enable zero-shot generalization—a process that's frequently complex; iii) there isn't

a universally adaptable system to address the $X$-Shot situation, where labels might vary in their visibility or frequency.

In this work, we leverage indirect supervision from an extensive assortment of NLP tasks. The Super-NaturalInstruction dataset (Wang et al., 2022) encompasses over 1,600 tasks across 76 categories. Each of these tasks is accompanied by instructions and numerous input-output examples. As depicted in Figure 2, this dataset offers an invaluable source of indirect supervision for our target $X$-Shot . For every task within the Super-NaturalInstruction dataset, we're presented with the associated instruction as well as (input, gold output) pairs. For each instance selected, we will randomly pick one output from the task label space that is different from the gold output, whether the task is generation or classification. As a result, we obtain one positive triplet (task instruction, input, gold output) and one negative triplet (task instruction, input, random output) for each example in our training dataset. Our indirect supervision stems from this dataset training. When evaluated on benchmark classification tasks, we convert every sample into triplets similarly, complemented by a human-written instruction. For an instance with text $t$ and $L_s$ positive labels, we add an instruction and craft $|L|$ triplets (task instruction, $t$, $l$) for each label $l$ from the label space $L$, with $L_s$ of them are positive and the remaining are negative.

Through this indirect supervision, minor alterations—be it a word or a few words—can pivot the class completely. By enabling the model to distinguish the positive and negative classes from marginally tweaked inputs, we ensure the model establishes more distinct decision boundaries.

**GPT Supervision for zero-shot labels.**   In addition to Instruction Supervision, we aim to enhance our model's performance on zero-shot labels. Given that we cannot procure annotated instances for these labels, how can we enhance the model's understanding of these labels without human intervention or labeling? This is where we leverage the capabilities of GPT (Brown et al., 2020) to produce weakly labeled examples. For generating instances related to zero-shot labels, we utilize in-context learning. This involves a random selection of demonstrations from either few-shot or frequently labeled data. Below is a sample prompt designed to generate entity typing text for a zero-shot type label:

```
entity type: paper
entity: New York Times
sentence:  I enjoy reading articles in The New York Times to stay
updated on current events and global news

entity type: gathering
entity:  concert
sentence: The concert was captivating, with the musicians' stellar
performance earning an encore request from the audience.

entity type: star
```

In this approach, upon exposing GPT to entity and entity statement examples associated with the entity type labels "paper" and "gathering", we introduce the zero-shot label "star". Subsequently, GPT generates an entity along with an entity statement, serving as a weakly supervised instance for this previously unseen label.

**Training strategy.**   We first train the RoBERTa (Liu et al., 2019) model on the transformed binary Super-NaturalInstruction dataset, then fine-tune on the augmented instances of downstream $X$-Shot tasks.

## 5   EXPERIMENTS

### 5.1   EXPERIMENTAL SETTING

**Datasets.**   Our objective is to choose datasets that can cover (i) multiple domains, (ii) various sizes of labels, and (iii) both single-label and multi-label scenarios. Therefore, we evaluate on three mainstream classification datasets: *FewRel* (Han et al., 2018), *UFET* (Choi et al., 2018), and

*Situation* (Yin et al., 2019), referring to relation exaction, entity typing, and situation identification problems respectively. All of them are considered the benchmark dataset in the text classification field. The number of labels in these datasets varies from 12 to 230 labels, making the classification task very challenging. In addition, an extra "*None*" label in one of the datasets makes the problem setting more realistic.

While the original datasets provide a foundation, they don't align with our needs because: i) some maintain consistent instance counts across all labels, whereas others display varied label coverage distributions; and ii) they aren't tailored for binary classification. To better accommodate the $X$-Shot scenario, we modify each dataset. This results in three distinct label groups: *freq-shot* labels, *few-shot* labels, and *zero-shot* labels. We'll delve into the specifics of this augmentation in the subsequent section.

• **FewRel (Han et al., 2018)** is a well-established relation classification dataset containing relation statements extracted by aligning terms from Wikipedia to the knowledge base facts in Wikidata. *FewRel* uses 64/16/20 relations for training, dev, and test sets, while each relation has 700 instances. Each instance in *FewRel* provides a relation statement, two entities from the sentence, and their corresponding relation label. Even though *FewRel* includes a large number of labels, the original experimental setting is to evaluate few-shot learning for a limited number of relations (Soares et al., 2019; Dong et al., 2020; Wang et al., 2020). Previous approaches usually perform an N-way K-shot learning, while N is usually 5 or 10 while K is usually 1 or 5.

To align with the objectives of $X$-Shot , our evaluation framework adopts a comprehensive setup, ensuring the inclusion of a diverse set of labels. Since the test set is not available for *FewRel*, we include 78 relations and divide them into 26/26/26 as freq/few/zero-shot labels. We put 500/5/0 instances for each freq/few/zero label in the training set, and 200 instances for each label in the dev/test set.

• **UFET (Choi et al., 2018)** is a human-labeled entity typing dataset with more than 5000 instances and 2519 unique labels. Each instance in *UFET* consists of an entity statement, the target entity, and the list of possible types of the entity. In contrast to *FewRel*, *UFET* is a free-form multi-label dataset while one instance can be labeled with several types of roles based on the context. UEFT has been studied as a multi-label classification problem in previous studies (Choi et al., 2018; Zhang et al., 2021).

For our approach, we adopted the most frequent 230 entity types and split them into 30/100/100 as the freq/few/zero-shot labels since the remaining entity type labels occur less than 20 times in the dataset. These 230 entities cover around 90% of the dataset. Since *UFET* is a multi-label dataset, it will be difficult to assign a specific number of instances per label. Therefore, we put all instances without zero-shot labels (around 70%) as the training set and the remaining as the dev/test set.

• **Situation (Yin et al., 2019)** is an event-typing classification dataset including 5,956 labeled instances. There are two kinds of situations here: i) 8 "need" situations where a specific kind of aid is needed, such as food or water supply. ii) 3 "issue" situations where an issue, such as a crime, is happening. Similar to *UFET*, this dataset is also a multi-label dataset. However, one thing that makes it different from the other datasets is that there is one special situation, "*None*", which means that none of the 11 situations fit. This dataset was used as a benchmark dataset for zero-shot classification in the previous study. The methodology is to convert it into a Natural Language Inference (NLI) binary classification problem, which is adopted as one of our baselines. If none of the labels are positive (receiving a probability higher than a threshold), then the "*None*" label is assigned.

To create a dataset with varying label occurrences, we separate the 11+1 situations as 4/4/4 freq/few/zero-shot labels, while the "*None*" label belongs to the zero-shot group. Similar to *UFET*, we treat instances without zero-shot labels (around 60%) as training instances and the remaining as the dev/test set.

For *UFET* and *Situation* datasets, even though we cannot assign a specific number of instances for each label in the training set due to the multi-label setting, we always limit the number of occurrences of few-shot labels to around 5 times in the training set in order to be consistent with *FewRel*. More dataset details are in Table 1.

**Baselines.** For baselines, we compare our system with the current state-of-the-art multi-way classification model, the in-context learning with GPT, and the most advanced few-shot/zero-shot learning methods in the literature.

Table 1: Dataset statistics

|          | domain        | #freq | #few | #zero |
|----------|---------------|-------|------|-------|
| FewRel   | Wikipedia     | 26    | 26   | 26    |
| UFET     | crowdsourcing | 30    | 100  | 100   |
| Situation | /            | 4     | 4    | 3+1   |

• **Multi-way classification (MWC, (Soares et al., 2019))** . In this baseline, we treat it as a traditional multi-way classification problem with a special Marker scheme called "Entity Marker"(Soares et al., 2019). Entity Marker introduces extra entity token markers to the model besides the entity terms and feeds the concatenation of start entity tokens into the classification head. For each statement containing entities, we put $< Ei >$ and $< /Ei >$ as the start and end entity tokens for each entity $i$. One example is as follows:

| `<E1>` LONDON `<\E1>` is the capital of `<E2>` UK `<\E2>` |
|---|

This methodology stands as the leading approach for extracting relation representations, especially in the realm of entity relation classification (Soares et al., 2019). We employ this strategy for both the *FewRel* and *UFET* datasets, given that they contain entities within their inputs. However, for the *Situation* dataset, given the absence of predefined entity spans in the situational statements, we continue to use the [cls] token as input for the classification head without integrating any Entity Markers.

• **Indirect Supervision from NLI (NLI, (Li et al., 2022))**. The previously established best approach for addressing a zero-shot classification challenge was to reframe it as an NLI task. This technique eliminates the need for specific annotations related to the label space or any label-specific data. A sequence classification task can be adapted into a text entailment problem by using the original statement as the premise and transforming the label into a hypothesis. Our method distinguishes itself from this NLI-centric technique in two significant ways: first, we broaden the range of indirect supervision sources from just NLI to encompass a diverse set of NLP tasks; second, we implement an instruction tuning schema rather than adopting a pairwise classification framework.

• **In-context learning with GPT (GPT-3.5).** For in-context learning, we create a prompt that includes three demonstrations, two positive and one negative, and each comes with the sentence, optional entity (entities), the relation/entity type/ situation term, and the label that indicates whether the term is correct. Then, we provide the same features for the instance we want to predict but let the GPT complete the label part. A template can be seen in Appendix A.1.

• **Prototypical Prompt learning (PPL, (Cui et al., 2022))** The most popular approach for addressing classification challenges within the few-shot framework is through the practice of prompt learning in recent years. It combines the strength of LLMs and a well-designed verbalizer that maps the model output to the pre-defined labels. This baseline utilized the prototypical verbalizer (ProtoVerb) that is built directly from training data N-shot setting converts classification into a sequence mask problem that, in each training iteration, the model puts N sentences from each label into a prompt and has the label token been replaced by the [mask] vector. For *FewRel*, *UFET*, and *Situation*, we select 500, 100, and 500 instances during training for prototype learning. Since we want to be consistent with the freq, few, and zero-shot learning approach, for freq and few shot labels, we keep selecting instances from the limited instances until we reach the number. For zero-shot labels, we simply put the label itself as the text for the training and test on the original test set.

**Implementation details** We elaborate on our implementation details at different stages here.

• **Indirect Supervision**. Consistent with the original experimental setup, we select 100 random instances from each task for training when compiling the indirect supervision dataset from SuperNaturalInstruction. Our prefix template follows the previous benchmark strategy, incorporating only the instruction and two positive examples—provided this inclusion doesn't surpass the word limit. When adjusting classification tasks to fit $X\text{-Shot}$, we draft three distinct instruction prompts and present the average outcomes to demonstrate the system's stability. Further details about each template are available in Appendix A.2.

• **GPT-3.5 for $D_{weak}$ collection.** We utilize the "text-davinci-003" GPT completion model for augmenting zero-shot instances. We configure the temperature to 1.6 to ensure more varied outputs and cap the maximum token output from GPT-3.5 at 80. However, GPT-3.5 doesn't always maximize this limit. For each zero-shot label, we generate 5 instances to serve as weak supervision.

Table 2: Main results on three benchmarks

| Models | FewRel | | | | UFET | | | | Situation | | | |
|---|---|---|---|---|---|---|---|---|---|---|---|---|
| | test | freq | few | zero | test | freq | few | zero | test | freq | few | zero |
| MWC (Soares et al., 2019) | 49.82 | 94.23 | 55.23 | 0 | 11.69 | 44.88 | 13.41 | 0 | 28.16 | 43.00 | 34.47 | 7.00 |
| NLI (Li et al., 2022) | 63.46 | **95.35** | 48.81 | 46.22 | 38.28 | 53.26 | 34.44 | **37.62** | 42.12 | **53.56** | 34.02 | 38.77 |
| PPL (Cui et al., 2022) | 53.23 | 95.15 | 63.54 | 0 | 3.28 | 10.63 | 3.48 | 0.89 | 25.37 | 22.83 | 26.78 | 26.48 |
| GPT 3.5 | 18.24 | 18.22 | 25.33 | 11.17 | 19.87 | 31.05 | 16.02 | 20.37 | **57.53** | 51.87 | **59.95** | **60.78** |
| $X$-Shot | **68.48** | 94.06 | 58.04 | **53.34** | **38.46** | **55.69** | **34.74** | 37.00 | 44.46 | 52.82 | 33.51 | 47.04 |

• **Prediction threshold.** Both NLI baseline and our method necessitate a threshold for assigning label predictions. We use the probability of the positive class produced by the model for this purpose. For *FewRel*, the label with the highest score is chosen. In *UFET* and *Situation*, we introduce a threshold parameter, $t$. Any label exceeding this probability threshold, $t$, is considered in the final prediction. We experiment with various values of $t$, ranging from 0.5 to 1, and select the optimal one. For *Situation*, there's a unique label "*None*" which signifies that none of the predefined situations are applicable. If no situation surpasses the threshold $t$, the label "*None*" is assigned.

## 5.2 RESULTS

The primary results are displayed in Table 2. Our model generally surpasses the baselines. While traditional multi-way classification excels with ample annotations, its performance falters in few-shot and especially zero-shot situations. Similarly, the few-shot prompting baseline struggles when encountering unseen instance texts, highlighting the constraints of classification models in the $X$-Shot context.

The in-context learning method shines in *Situation* with its limited 12 labels and simpler nature. However, $X$-Shot still exceeds all other baselines significantly. Also, when it comes to the other two datasets where we have hundreds of labels, the model can no longer make wise decisions.

While the NLI-based indirect supervision—a prevalent method that transforms the zero-shot task into an existing NLI problem—delivers impressive results across various settings, our method proves to be even more potent. This underscores the superior robustness of the instruction-learning approach in the context of the $X$-Shot setting.

### 5.2.1 ANALYSIS

**Error Analysis.** To analyze the error patterns, we pick *Situation* dataset as the example and collect the most typical errors as follows:

• **Bias toward more frequent labels** Under our multi-label classification problem setting where the number of labels can be up to 230, it would be very common for multiple labels to have similar semantic meanings. Even with the situation dataset where we have the least number of labels (11+1), we can still find similar labels, such as "terrorism" versus "crime/violence". For example, one input sentence from the Situation dataset is "@-@ Maiduguri hit @-@ with boko haram squeezed out of captured territory, security analysts have predicted a rise in bomb attacks in towns and cities, including to disrupt elections in three weeks ' time." Even though the gold label is "terrorism", it gives 0.99 probability for "crime/violence" considering it does have a similar meaning and as a frequent label it has been seen multiple times. We can see that the frequency of the label being seen can be an important factor, especially since we have a massive amount of labels that can easily confuse the model.

• **Misled by Textual Cues** Occasionally, the input sentence includes terms directly related to one of the labels, even if the context doesn't correspond to that label.

For example, one input sentence is "the two dead adults were either villagers or rescuers searching for those missing, xinhua added" include, which mentions the term "rescuers". The model strongly favors the "search/rescue" label, while none of the labels fit and the gold label is actually " None".

8

• **Ambiguous labels** It's common for people to have disagreements on the annotations. Sometimes the model makes a more appropriate judgment than the data provides to some people's perspectives. One such example is "so much untreated sewage has been pumped into sierra leone 's rivers and coastal waters that much of the water itself is contaminated with the cholera bacteria , unicef said .". The ground truth label for this input is "utilities, energy, or sanitation". However, the model also strongly suggested "medical assistance", a fair choice given the mention of "cholera bacteria".

**Why do few-shot labels outperform zero-shot labels at times?**     We observe that within the UFET and Situation benchmarks, the performance of few-shot labels is slightly worse than zero-shot labels. We hypothesize that this outcome is attributed to the robustness of LLMs when endowed with extensive pretrained knowledge, wherein both scenarios of no fine-tuning and fine-tuning with ample data manifest resilience. Conversely, minimal fine-tuning tends to induce overfitting.

**Influence of Task Type Overlap.**    The Super-NaturalInstruction dataset doesn't directly include our target datasets. We removed the top 10 tasks closest to each test dataset to assess the impact of similar tasks. The measurement is based on cosine similarity between Sentence-BERT embeddings of the 757 task definitions in the Super-NaturalInstruction dataset and each test dataset's instruction.

Table 3: Results of retraining the model after deleting top-10 similar tasks

|           | test  | freq  | few   | zero  |
|-----------|-------|-------|-------|-------|
| FewRel    | 63.34 | 89.04 | 60.95 | 40.04 |
| UFET      | 38.05 | 53.39 | 34.20 | 37.30 |
| Situation | 41.96 | 49.76 | 36.96 | 39.15 |

Comparing results in Table 3 with those in Table 2, there's a minor performance decline for FewRel and Situation datasets. However, UFET's performance remains stable. This suggests that similar tasks in the Super-NaturalInstruction dataset can be beneficial. Even with slight decreases, results still surpass baseline levels, underscoring the value of diverse training tasks. This is further supported by subsequent analysis.

**Number of Tasks vs Number of Instances.**    Balancing the number of tasks and the number of instances per task is pivotal in data collection. We wonder, by keeping the total instance count constant, should we have more tasks or more instances per task? We try [100,200,..,700] for the varying number of tasks, each with 100 instances.

In total, we have [10,000, 20,000, ... 70,000] instances. Accordingly, for the varying number of instances per task, we have datasets with [10,000/757, 20,000/757, ... 70,000/757] number of instances. The overall instances remain the same in each step.

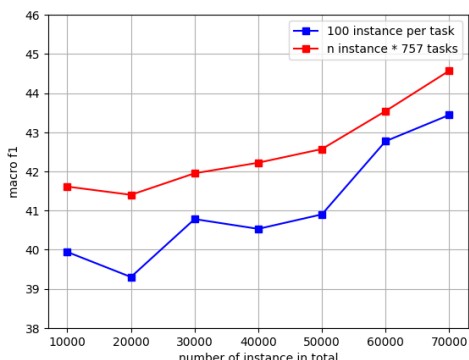

Figure 3: #instances vs. #tasks

From Figure 3, it's evident that both task count and instance count boost performance. While increasing either is beneficial, having more tasks has a greater impact than adding more instances to each task. Given these insights, future work should focus on diversifying the types of tasks exposed to the model, considering data constraints.

## 6    CONCLUSION

This work introduces $X$-`Shot` , a challenging text classification framework where labels range from non-existent to frequent. $X$-`Shot` reflects realistic scenarios where we encounter frequent-shot, few-shot, and zero-shot labels simultaneously. Our innovative approach recasts any text classification issue into a binary task, handling varying label amounts and frequencies. We introduce $X$-`Shot` to navigate this intricate challenge, leveraging instruction learning and PLMs' weak supervision. Our approach consistently outperforms the latest methods across three benchmark datasets in both single and multi-label contexts.

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

# A APPENDIX

## A.1 IN-CONTEXT LEARNING TEMPLATE

For the in-context learning baseline, we provide 3 demonstrations, 2 positive ones and 1 negative one, and let GPT complete the label of the test instance. The template is as follows:

```
Sentence:        Pan was appointed director of the National Academy
(Zhejiang Academy of Fine Arts) by the Kuomintang Ministers
Entity 1: Chen Lifu
Entity 2: Kuomintang
Relation: member of political party
Label: Yes

Sentence:        Aldo Protti (July 19 ,1920 - August 10 , 1995 ) was an
Italian baritone opera singer
Entity 1: Aldo Protti
Entity 2: baritone
Relation: voice type
Label: Yes

Sentence:         Part of DirectXDirect3D is used to render three -
dimensional graphics in applications
Entity 1: DirectX
Entity 2: Direct3D
Relation: movement
Label: No

Sentence:  The Suzuki GS500 is an entry level motorcycle manufactured
and marketed by the Suzuki Motor Corporation.
Entity 1: Suzuki GS500
Entity 2: Suzuki Motor Corporation
Relation: winner
Label:
```

## A.2 TASK INSTUCTIONS

To prove the robustness of our model, we create 3 versions of the task instructions for each of the datasets (*FewRel*,*UFET*,*Situation*) as follows:

*FewRel*
Instruction A:    Given a sentence about two entities, return a relation between the two entities that can be inferred from the sentence.
Instruction B:     Your task is to identify a relationship between two entities mentioned in a given sentence.
Instruction C:     Identify the relationship between two entities in a given sentence that can be inferred from the sentence.

*UEFT*
Instruction A:     Given a sentence about an entity, return the type of entity that can be inferred from the sentence.
Instruction B:  The task is to identify the type of an entity mentioned in the sentence based on the information provided in the sentence.
Instruction C:     Determine the type of entity mentioned in the given sentence by analyzing the context of the sentence.

*Situation*
Instruction A:  Given a sentence about a situation, return the type of the situation that can be inferred from the sentence.
Instruction B:   The task is to identify the situation mentioned in the sentence based on the information provided in the sentence.
Instruction C:  Determine the situation mentioned in the given sentence by analyzing the context of the sentence.

