# OpenReview forum: "X-SHOT: A Single System to Handle Frequent, Few-shot and Zero-shot Labels in Classification"
_ICLR.cc/2024/Conference — ICLR 2024 Conference Withdrawn Submission_

### Official Review · Reviewer_mNFM · 2023-10-25

**Soundness:** 3 good
**Presentation:** 3 good
**Contribution:** 3 good
**Rating:** 5
**Confidence:** 3

**Summary:**

The paper proposes a unified framework for X-Shot learning problem. It is more practically useful since a mixture of situations can often exist in practice. To address such a novel problem, instruction learning is employed wiht the help of augmented data generated by pre-trained language models. Empirical studies demonstrate the proposed unified system surpasses SOTA techniques on three benchmark datasets.

**Strengths:**

-- The problem X-shot learning is a novel one and it is practically useful.

-- The presentation is generally good with comprehensive introduction of background and related works.

**Weaknesses:**

-- In section 4.1, it is unclear what the instruction term stand for. It would be helpful to give some examples in the text or Figure 1 for clarity.

-- It is also unclear how the "indirect supervision" works. Again, including concrete examples in Figure 2 would be helpful.

-- The experimental results do not provide convincing proof that the proposed approach outperform its counterparts. The comparison may be unfair since varying resources have been used in different methods.

**Questions:**

1. How is the computational complexity of the proposed method?
2. Is it possible to incorporate existing SOTA methods for zero/few-shot learning methods into the proposed framework? Would this further improve the performance?
3. Can I understand the proposed approach as a training data augmentation method? If yes, it might be inaccurate to treat it as a zero/few-shot learning algorithm. Please clarify this point.

---

### Official Review · Reviewer_WJCJ · 2023-10-30

**Soundness:** 2 fair
**Presentation:** 3 good
**Contribution:** 3 good
**Rating:** 6
**Confidence:** 3

**Summary:**

The paper introduces X-SHOT, a unified system for text classification tasks such as relation classification, ultra-fine entity typing, and situation detection that can handle any label occurrence (frequent-shot, few-shot, zero-shot). According to the authors, X-SHOT leverages Instruction Learning, which involves using data autonomously generated by pre-trained Language Models (PLMs) to learn task instructions. These instructions are then used to guide the classification of inputs into labels. The paper demonstrates that X-SHOT outperforms preceding state-of-the-art techniques on three benchmark datasets across diverse domains in both single-label and multi-label classifications.

**Strengths:**

Originality: The paper introduces a new problem setting for text classification, X-SHOT, which can handle any label occurrence, whether it be frequent-shot, few-shot, or zero-shot. This is a novel problem formulation according to the authors. But since I am not well-versed in this line of work, I am not confident about the authors's claim on the originality. I encourage other reviewers comment on this point. The paper reframes text classification tasks into a binary classification task, adaptable to any number of labels and occurrences. I think it is a neat approach that unifies various text classification tasks into a single framework. It is something I have not read about before, but again I am not an expert in this area, so please refer to other reviewers' comments on this point.

Quality: The paper provides a thorough evaluation of X-SHOT on three benchmark datasets across diverse domains in both single-label and multi-label classifications. The evaluation demonstrates that X-SHOT outperforms preceding state-of-the-art techniques in FewRel and UFET.

Clarity: The paper is well-organized and easy to follow, with clear section headings and subheadings. The paper provides detailed explanations of technical terms and concepts, making it accessible to readers with varying levels of expertise.

Significance: The paper's contributions seems to have significant implications for the field of text classification, as X-SHOT provides a unified system that can handle any label occurrence, making it a versatile solution for real-world applications.

**Weaknesses:**

* May be over claiming contributions: I have to admit that I have not read a lot of papers in few-shot/zero-shot/one-shot domain. It seems quite unexpected to me that no one had ever attempted to develop a system that can utilize labels of all frequencies. After reading this paper, I did some simple google search. Without going into too much details, I found some articles discussing models that can handle "N-shot" labels with N ranging from 0 to some N, albeit N is not claimed to be set to infinity, which seems to be very similar to the "X-shot" problem formation. Since I am no expert in this topic and I certainly do not intend to become on for the purpose of this review, I encourage other reviewers who have more domain knowledge than me to think about this point.
* The fact that the proposed model does not do well on the Situation dataset can be problematic: From table 2 we can see that the proposed model does not do well on the Situation dataset and it certainly did not "still exceed[s] all other baselines significantly." In particular, I find it hard to be persuaded that the proposed model exceeds NLI (Li et al., 2022) **significantly**. Can authors elaborate on this?
* Other aspect might worth considerations yet unexplored: For example, for the Situation dataset, GPT3.5 is able to outperform all other models. However, the authors could have argue that GPT3.5 might be less advantageous because of computational efficiency. The paper does not provide a detailed comparison of X-SHOT with other state-of-the-art techniques in terms of computational efficiency. It is unclear how X-SHOT compares in terms of training time and computational resources required.

**Questions:**

* Can you provide more details on the Instruction Learning technique used in X-SHOT? Specifically, how is the data autonomously generated by pre-trained Language Models (PLMs) used to learn task instructions?
* Can you provide more details on the hyperparameters used in the experiments? Specifically, how were the hyperparameters selected and what is the impact of different hyperparameters on X-SHOT's performance?
* Can you explain why you choose text classification tasks over other tasks for experiments and the possibility of adapting your proposed framework to other classification settings (e.g. image classification)?

---

### Official Review · Reviewer_GeK1 · 2023-11-02

**Soundness:** 2 fair
**Presentation:** 2 fair
**Contribution:** 2 fair
**Rating:** 5
**Confidence:** 4

**Summary:**

The authors introduce a new classification challenge that data with zero-shot, few-shot and frequent-shot labels can be trained without predefined limits. To solve this challenge, they propose a unified training framework X-shot, which seamlessly transitions most classification challenges into a unified, instruction-driven binary classification task. They conduct experiments on three benchmark datasets across diverse domains in both single-label and multi-label classifications to prove the effectiveness of their proposed method.

**Strengths:**

1. The authors propose an interesting challenge that the zero-shot, few-shot and frequent-shot tasks can be trained within one unified framework, which has never been studied before.
2. The way to leverage the indirect supervision by enabling the model to distinguish the positive and negative classes from marginally tweaked inputs is worth thinking.

**Weaknesses:**

1. The proposed method lacks novelty. It is unclear which part is more important. If the pretraining stage on Super-NaturalInstruction Dataset is more critical, it seems unfair to directly compare with those competitor methods which adopts no pretraining stage.
2. The results seems not good enough, especially on the Situation benchmark.
3. In order to prove the practical value of the proposed X-shot that the zero-shot, few-shot, and frequent-shot problems can be well-solved by the unified training, I think the final results should be compared with the additional results retrieved by disjoint training. For example, the zero-shot results retrieved by unify training should be better than that retrieved by only training on all zero-shot tasks.

**Questions:**

See my comments in Weaknesses.

---

### Official Review · Reviewer_hFW1 · 2023-11-08

**Soundness:** 3 good
**Presentation:** 4 excellent
**Contribution:** 2 fair
**Rating:** 3
**Confidence:** 3

**Summary:**

The paper introduces a novel problem setting, *X-shot*, which can perform multi-label classification encompassing freq-shot, few-shot, and zero-shot labels. To achieve this, the paper transforms the multi-label classification into label-wise binary classifications, with input triples of <instruction, input, label>. The paper also addresses the problem of limited annotation by incorporating indirect supervision as a knowledge source and leveraging GPT supervision to enhance the performance of zero-shot labels. The method is validated across three datasets (including FewRel, UFET, and Situation), and yields state-of-the-art results.

**Strengths:**

1. The paper is well-written and easy to follow.
2. The paper unifies freq-shot, few-shot, and zero-shot labels in the proposed *X-shot* setting.
3. The ideas of indirect supervision and GPT-supervision are reasonable.

**Weaknesses:**

1. The solution for *X-shot* setting is not new (Sec 4.1). Converting multi-label classification into binary classifications is a conventional approach in ML algorithms.
2. While GPT supervision is proposed for zero-shot labels (Sec 4.2), there is no ablation study on the influence of generated instances on freq-shot, few-shot, and zero-shot labels.
3. The improvement looks marginal compared to previous works (Tab. 2).

**Questions:**

1. There should be a paragraph in Related Work (Sec 2) discussing multi-label or multi-class classifications.
2. Since GPT supervision is used for zero-shot labels, the results of *X-shot* are much higher than that of GPT itself (Tab. 2). Could the authors provide more explanations and analysis on where the improvement comes from (especially on the zero-shot labels of FewRel and UFET datasets)?
3. What is the efficiency of *X-shot*? Is *X-shot* using less supervision or is more time-efficient than *NLI*?